# Impact of Bee and Fly Pollination on Physical and Biochemical Properties of Strawberry Fruit

**Muhammad Anees [1], Mudssar Ali [1,\*], Hamed A. Ghramh [2,3,4], Asif Sajjad [5], Khalid Ali Khan [2,3,6,\*], Shafqat Saeed [1] and Kashif Razzaq [7]**

[1] Institute of Plant Protection, Muhammad Nawaz Shareef University of Agriculture, Multan P.O. Box 60000, Pakistan

[2] Research Center for Advanced Materials Science (RCAMS), King Khalid University, P.O. Box 9004, Abha 61413, Saudi Arabia

[3] Unit of Bee Research and Honey Production, King Khalid University, P.O. Box 9004, Abha 61413, Saudi Arabia

[4] Biology Department Faculty of Science, King Khalid University, P.O. Box 9004, Abha 61413, Saudi Arabia

[5] Department of Entomology, Faculty of Agriculture and Environment, The Islamia University of Bahawalpur, Bahawalpur P.O. Box 63100, Pakistan

[6] Applied College, King Khalid University, P.O. Box 9004, Abha 61413, Saudi Arabia

[7] Department of Horticulture, MNS-University of Agriculture, Multan P.O. Box 60000, Pakistan

\* Correspondence: mudssar.ali@mnsuam.edu.pk (M.A.); khalidtalpur@hotmail.com (K.A.K); Tel.: +92-303-0770112 (M.A.)

**Abstract:** Improvement in physical and biochemical properties of fruits through cross-pollination is a highly variable phenomenon. It mainly depends on the species of the pollinator and the nature of the crop being pollinated. It is therefore imperative to quantify the relative pollination effectiveness of an insect species for a certain crop species. In the present study, pollination effectiveness of two native bees (i.e., *Apis dorsata* and *A. florea*) and two syrphid flies (i.e., *Eristalinus aeneus* and *Eupeodes corollae*) were evaluated in terms of physical properties of strawberry fruits at Multan, Pakistan. The physical parameters of resultant fruits included length, pole, equator, fruit set ratio, number of days to reach market maturity, and shelf life. The biochemical properties of fruits resulted from open-pollinated plants (free insect visits) and caged plants (no insect visits) were also compared. The biochemical parameters included TSS (total soluble solids), TA (titratable acidity), vitamin C, and pH. *Apis dorsata* was the most abundant pollinator, followed by *E. aeneus* and *E. corollae*. Based on single-visit effectiveness, *A. dorsata* proved to be the most effective pollinator, in terms of physical properties, of strawberry fruit. *Eristalinus aeneus* outcrossed *A. florea* in terms of fruit set (%). The open-pollinated plants showed better physical and biochemical properties (26% higher TSS, 34% higher TA, but 25% lower pH value) in terms of fruits than the caged plants. Conservation of *A. dorsata* and *E. aeneus* can enhance physical and biochemical properties of strawberry fruits in the region.

**Keywords:** foraging behavior; single-visit efficiency; open-pollination; market maturity; shelf life

## 1. Introduction

Strawberries are aggregate fruits as each floral receptacle consists of multiple carpels [1] and cross-pollination can increase the number of fertilized achenes by 62% [2]. At least 70–80% of carpels must be visited by the insect pollinators for proper fruit set [3]. Poor fertilization is therefore the main reason for malformation in strawberry fruits [4].

Because of its generalized floral characteristics (i.e., radial symmetry, disc-shaped flowers with approachable nectar and exposed anthers), strawberry flowers are visited by a wide array of pollinating insects [5,6]. To obtain a good strawberry yield, as well as managed honey bees, a diverse assemblage of native pollinator species is required [5–7]. A recent study [8] has reported wild bees as superior pollinators of strawberry plants over European honey bees in terms of fruit weight. Hoverflies e.g., *Episyrphus balteatus* and

*Eupeodes latifasciatus*, can also serve as supplementary pollinators, improving fruit set by 58% [9,10].

Pollination may impact the shelf life of strawberries in terms of metabolic processes [11]. More than 90% of fruits can become non-marketable after only 4 days in storage [12]. Little is known about the effects of cross-pollination on the fruit quality and nutrient contents of strawberry fruits. Some recent studies have reported a positive impact of insect pollination on physical and biochemical properties of strawberry fruits, i.e., up to 90% improvement in the commercial value of marketable fruits [5,10,13,14]. Open-pollinated fruits exhibit high TSS [15], high brix value [2,10], and low acidity [15]. They also exhibit high auxin concentrations in strawberry achenes, which is mainly responsible for promoting fruit enlargement [14,16].

Additional to biochemical properties, the physical properties are also significantly enhanced by cross-pollination in strawberry fruits in terms of weight, color, size, shape, commercial value, and post-harvest quality [13,15]. The shelf life of fruits are usually sorted into commercial grades on the basis of aberrations in shape (deformations), color (areas with yellow or green color), firmness, and size (fruit diameter) [13,17].

Pollinator species vary in their effectiveness on the basis of single-visit efficiency in terms of pollen harvest [18,19], pollen deposition on stigma [18,20,21], and, more importantly, seed or fruit set [22–24]. Some previous studies from the southern region of Punjab, Pakistan, have reported bees and syrphid flies as the most effective pollinators of different crops in terms of their single-visit efficacy [18,19,25,26].

Previously, no study has evaluated the effectiveness of native bees and syrphid flies in strawberry pollination for Pakistan or worldwide. We found only a single study that evaluated the effectiveness of insect pollinators in terms of single-visit efficacy in strawberry plants. Therefore, the current study was planned to evaluate the single-visit pollination effectiveness of honey bees and syrphid flies in terms of pollen harvest and fruit set efficacy of strawberry crops. Self- and open-pollinated treatments were also maintained in order to determine the impact of unrestricted insect visits on the physical and biochemical properties of strawberry fruits.

## 2. Materials and Methods

### 2.1. Study Area

Experiments were conducted in a farmer's field 30.2887° N, 71.5279° E (Bun Bosan, Multan). The owner of the field granted permission to conduct trials in his field (Acknowledgements). Strawberries (var. Chandler) were grown on an area of 0.5 ha in double rows on 0.457 m wide beds with plant to plant distance of 6 inches during the last week of October, 2018–2019. The strawberry runners were purchased from a commercial nursey located at Dir (35.1977° N, 71.8749° E), a district of Swat Khyber Pakhtunkhwa, Pakistan. The study area is arid and experiences hot summers and cold winters. The average temperature in summer is 40 ± 5 °C, and in winter it is 10 ± 5 °C; moreover, yearly total rainfall ranges from 127–254 mm [27].

### 2.2. Pollinator Abundance

Data regarding pollinators' abundance were recorded twice a day during peak activity hours (i.e., 9:00 am and 12:00 pm), with 3-day intervals from mid-February to mid-April 2021 (i.e., total 30 censuses). During each census, linear transect walks were performed for 120 min along 10 randomly selected beds (60 m each) to count all the insect pollinators visiting strawberry flowers. The insects were initially morphotyped during systematic observations and identified to the lowest taxonomic level later on (Acknowledgements).

### 2.3. Foraging Behavior

The foraging behavior of the top four most abundant pollinators (having >35 individuals) was recoded, in terms of single-visit pollen harvest, visitation rate (number of flowers visited/minute), and stay time (time spent by an insect/flower), using a stopwatch. To

measure pollen harvest during a single visit, several floral buds were caged before they opened and un-caged after they had opened fully. Only one individual of a particular species was allowed to visit a plant, and was subsequently caught in order to count the number of pollen grains attached on its body (N = 20 per pollinator species) following the procedure of [28]. This procedure was performed early in the morning, when pollinators just began their foraging activity, thereby diminishing overestimation in pollen count. Their affinity for nectar and/or pollen was also recorded by careful visual observation. The foraging behavior of pollinators was observed by two different observers across the peak flowering period of mid-February to mid-March.

### 2.4. Single-Visit Effectiveness

In order to determine the single-visit effectiveness of insect pollinator species, 20 floral buds were caged (with nylon mesh bags to allow only air to pass and not the small pollinator species) for each individual species prior to their opening, and were un-caged on the next day once they were fully open. One individual of a particular pollinator species was allowed to visit a flower, and the flower was re-caged until senescence. Single-visit efficacy was evaluated in terms of resultant fruit set percentage, fruit weight (g), fruit length (length), and fruit width (pole and equator). The single-visit effectiveness of pollinators was measured by two different observers across the peak flowering period of mid-February to mid-March.

### 2.5. Physical Parameters

For measuring the reproductive success, 100 plants for each group, open-pollinated and self-pollinated, were maintained for the comparison. Fruits were harvested at similar maturity stage (75% ripening), and the following physical parameters were recorded: fruit weight (g), fruit length (length), fruit width (pole and equator), fruit set percentage, number of days to reach market maturity (75% ripening), and shelf life (up to 7 days).

To compare the effect of pollination treatment (open- and self-pollinated plants) on fruit shelf life, 100 fruits were selected from both treatments of roughly equal size and weight. Fruits were kept at room temperature and observed twice a day (at 09:00 am. and 16:00 pm local time). At each observation hour, unmarketable fruits were removed from treatments and remaining fruits were weighed using an electronic weighing balance. We considered a fruit unmarketable if it showed 5 to 10% decay and 10–25% shriveling. Moreover, marketable fruit percentage was determined using following formula:

$$\text{Marketable fruit} = \frac{\text{Total no. of fruits} - \text{Decayed fruit removed}}{\text{Total no. of fruits}} \times 100$$

### 2.6. Biochemical Parameters

To evaluate the impact of pollination on the physiochemical properties of both treatments, the following parameters were recorded from the extracted juice of marketable strawberry fruit: total soluble solids (TSS), titratable acidity (TA), pH, and vitamin C [29]. Abbe's refractometer was used to calculate the TSS of the strawberry samples. The apparatus was standardized with purified water and adjusted to 40 °C. The lens was cleaned with toluene, and then 2–3 drops of strawberry juice were deposited onto the lens and the reading was noted.

TA of strawberry juice was measured as per the method of [30]. First, 10 mL of strawberry juice was transferred into a 100 mL conical flask and distilled up to 50 mL with distilled water. Subsequently, it was titrated against 0.1 N sodium hydroxide using 2–3 drops of phenolphthalein as an indicator until the end point, i.e., pink coloration, was attained. The pH was calculated via a pH meter. The bulb of the pH meter was dipped in 10 mL strawberry juice, and the reading on the screen was noted. Three replications of each treatment were performed.

Vitamin C content of the juice was determined following the method of [31]. In brief, 10 mL of strawberry juice was transferred into a 100 mL volumetric flask and the volume

was made up by adding 0.4% oxalic acid solution. From this, 5 mL of filtrated aliquot was taken, and this was titrated against 2, 6-dichlorophenolindophenol dye until the end point, i.e., light pink coloration, was reached (persisted at least for 15 s).

*2.7. Data Analysis*

The data regarding stay time, visitation rate, pollen harvest, and single-visit efficiency (in terms of length, pole, equator, weight, and fruit set percentage) of honey bees and syrphid flies was analyzed using analysis of variance (ANOVA). Means were compared via Tukey's test at $p = 0.05$. Mann–Whitney U test was applied (as data did not follow a normal distribution) to compare open-pollination and self-pollination treatments in terms of physical parameters (fruit length, fruit pole, fruit equator, fruit weight, fruit set percentage, and number of days to reach market maturity), an independent sample *t*-test was applied to compare the means of open-pollination and self-pollination treatments in terms of biochemical parameters (TSS, T.A, vitamin C, and pH). All statistical analysis was performed using XLSTAT software.

**3. Results**

The floral visitor community of the strawberry crop was composed of seven bee and five syrphid fly species (Table 1). However, *Xylocopa* sp., *Nomia* sp., *Colelioxys* sp., *Amegilla* sp., and *Thyreus* sp. were rarely seen, and did not come under our systematic data recording criteria. Among all floral visitors, *Apis dorsata* was the most abundant, followed by *Eristalinus aeneus* and *Eupeodes corollae*, while lowest abundance was recorded for *Ischiodon scutellaris*. The abundance of syrphid flies (56%) was higher than of bees (44%). Both the honey bee species foraged for nectar. Three syrphid species (*E. aeneus*, *E. megacephalus*, and *I. scutellaris*) fed on both nectar and pollen, while two species (*E. corollae* and *Episyrphus balteatus*) solely fed on pollen (Table 1).

**Table 1.** Insect species feeding on strawberry flowers, with their total abundance, relative proportion, and foraging task in a strawberry field at Multan, Pakistan, from February to April, 2021.

| Order | Family | Genus/Species | Total Abundance | Relative Proportion % | Foraging Task (N/P) |
|---|---|---|---|---|---|
| Hymenoptera | Apidae | Apis dorsata | 81 | 30.22 | N |
| | | *Apis florea* | 35 | 13.06 | N |
| Diptera | Syrphidae | *Eristalinus aeneus* | 67 | 25.0 | N/P |
| | | *Eupeodes corollae* | 39 | 14.55 | P |
| | | *Episyrphus balteatus* | 19 | 7.09 | P |
| | | *Eristalinus megacephalus* | 15 | 5.59 | N/P |
| | | *Ischiodon scutellaris* | 12 | 4.48 | N/P |
| | | Total syrphid flies | 152 | 56.71 | |
| | | Total honey bees | 116 | 43.28 | |

N, nectar; P, pollen.

Significant differences were observed among the abundant pollinator species in terms of stay time (F = 9.32, d.f. = 3, $p < 0.000$) and visitation rate (F = 14.63, d.f. = 3, $p < 0.000$). Stay time was the highest for *E. corollae*, while it was the lowest for *A. dorsata*. Based on visitation rate, *A. dorsata* visited the highest number of flowers, followed by *E. aeneus*, while the lowest number were visited by *E. corollae* and *A. florea*. (Table 2). All four pollinators differed in terms of pollen harvest, since the median line of each box plot lies outside the boxes of the comparison box plots. Although the data is highly scattered and more skewed in *A. dorsata*, pollen harvest was still far greater than for the other three pollinators. *Eristalinus aeneus* was the second most efficient pollen harvester after *A. dorsata*; although the distribution is skewed, it was considerably less scattered (Figure 1).

**Table 2.** Foraging behavior of two honey bee and two syrphid fly species in terms of stay time and visitation rate in a strawberry field in Multan, Pakistan.

| Pollinator Species | Stay Time/Flower/Visit (N = 100) | Visitation Rate (No. of Flowers Visited/Min) (N = 100) |
|---|---|---|
| *A. dorsata* | 3.70 ± 0.19b | 9.69 ± 0.27a |
| *A. florea* | 14.46 ± 1.08a | 2.60 ± 0.22b |
| *E. aeneus* | 6.90 ± 1.02b | 7.66 ± 0.34a |
| *E. corollae* | 15.08 ± 9.83a | 2.50 ± 0.22b |

Mean values sharing similar letters in respective columns show non-significant differences according to Tukey's test at the 5% level (±SE). N = number of observation.

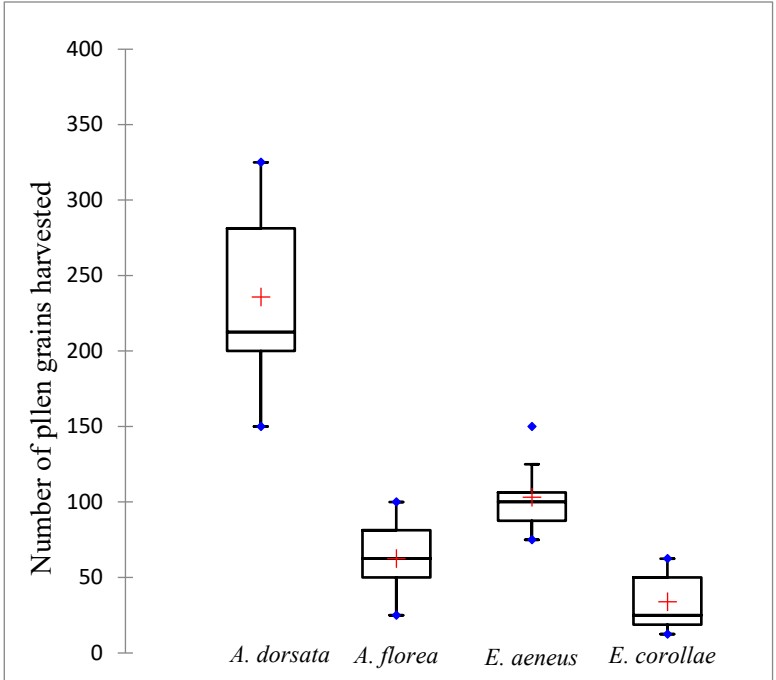

**Figure 1.** Box-and-whisker plot of single-visit pollen harvest of insect pollinators.

There was also a significant difference among pollinator species in terms of their single-visit effectiveness, i.e., fruit length (F = 48.62, d.f. = 3, $p < 0.000$), pole (F = 1.78, d.f. = 3, $p < 0.0$), equator (F = 5.22, d.f. = 3, $p < 0.010$), fruit weight (F = 60.21, d.f. = 3, $p < 0.000$), and fruit set ratio (F = 41.3, d.f. = 3, $p < 0.000$). The results reveal that *A. dorsata* was the best pollinator in terms of single-visit fruit length, fruit diameter (pole and equator), fruit weight, and fruit set (%), followed by *E. aeneus* (Table 3).

**Table 3.** Fruit length, pole, equator, weight, and fruit set resulting from single visits by the two honey bee species and two syrphid fly species.

| Pollinator Species | Length (cm) | Pole (cm) | Equator (cm) | Weight (g) | Fruit Set (%) |
|---|---|---|---|---|---|
| *A. dorsata* | 22.23 ± 1.01a | 9.24 ± 2.02a | 7.11 ± 0.7a | 19.34 ± 1.67a | 80a |
| *A. florea* | 6.06 ± 0.78c | 7.88 ± 2.83a | 4.60 ± 0.92b | 8.37 ± 2.4b | 40c |
| *E. aeneus* | 12.57 ± 0.26b | 6.39 ± 0.84b | 4.06 ± 2.11b | 7.37 ± 1.28b | 60b |
| *E. corollae* | 4.02 ± 0.67c | 4.17 ± 2.94b | 1.89 ± 0.76c | 6.01 ± 0.43b | 30d |

Mean values sharing similar letters in respective columns show non-significant differences according to Tukey's test at the 5% level (±SE).

The Mann–Whitney U test revealed a significant difference between open-pollinated and self-pollinated treatments in terms of physical parameters of strawberry fruits, i.e.,

fruit length, diameter (including pole and equator), fruit weight, number of days to reach market maturity, and fruit set percentage. The results reveal a 62% increase in fruit length, 50% in fruit weight, 26% in fruit set percentage, and 3 less days to reach market maturity in open-pollinated fruits as compared to self-pollinated fruits (Table 4). The open-pollinated fruits showed better shelf life (7 days) and marketable percentage (100% on 3rd day) than self-pollinated fruits (4 days and 56% on 3rd day) (Figure 2).

**Table 4.** Comparison of mean of ranks of physical parameters of strawberry fruits between open- and self-pollination treatments.

| Results of Mann–Whitney U Test | Length (cm) | Pole (cm) | Equator (cm) | Weight (g) | Days to Reach Market Maturity | Fruit Set (%) |
|---|---|---|---|---|---|---|
| Mean of Ranks (open-pollination) | 32.745 | 28.745 | 28.138 | 31.809 | 8.871 | 22.25 |
| Mean of Ranks (self-pollination) | 14.755 | 18.755 | 19.362 | 15.691 | 22.629 | 18.25 |
| Mann–Whitney U | 306 | 682 | 739 | 394 | 54 | 640 |
| *p*-value | 0.0001 | 0.0011 | 0.0048 | 0.0001 | 0.0001 | 0.0253 |

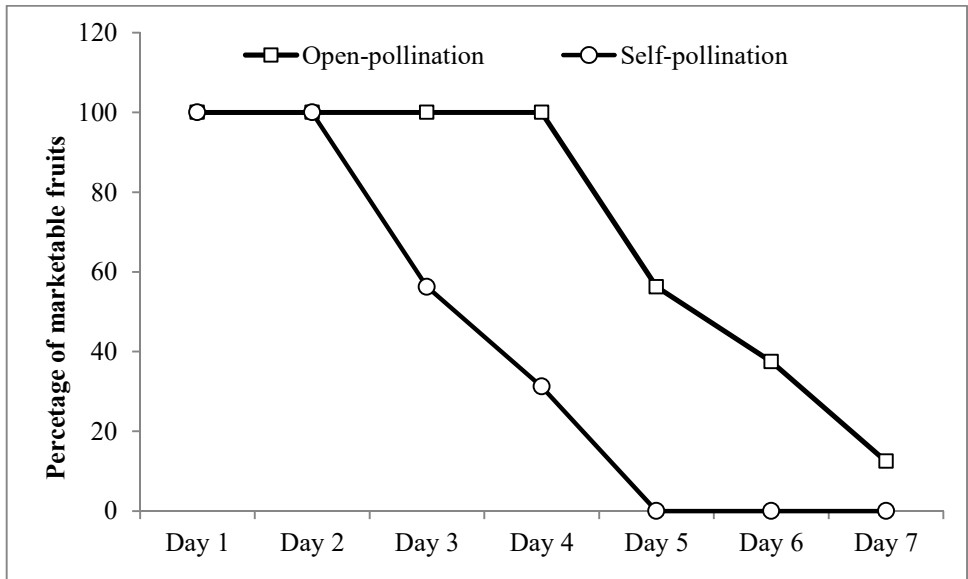

**Figure 2.** Percentage of marketable fruits with increasing number of days after harvesting in open- and self-pollinated fruits.

The *t*-test revealed significant differences between open-pollination and self-pollination in terms of biochemical parameters of strawberry fruits, i.e., TSS ($p < 0.000$), TA ($p < 0.03$), and pH ($p < 0.000$), while vitamin C did not differ significantly between the treatments (0.484). The open-pollinated fruits showed 26% higher TSS, 34% higher TA, but a 25% lower pH value than self-pollinated fruits (Table 5).

**Table 5.** Comparison of means (±SE) of chemical parameters of strawberry fruits between open- and self-pollination treatments.

| Pollination | TSS (Degree Brix) | TA (%) | Vitamin C | pH |
|---|---|---|---|---|
| Open-pollination | 7.14 ± 0.07 | 0.39 ± 0.03 | 220.6 ± 10.16 | 2.82 ± 0.11 |
| Self-pollination | 5.64 ± 0.25 | 0.29 ± 0.04 | 208.3 ± 13.17 | 3.79 ± 0.16 |
| Results of *t*-test | | | | |
| *t*-observed | 5.622 | 2.023 | 0.740 | −4.718 |
| *t*-critical | 2.074 | 2.074 | 2.074 | 2.074 |
| *p*-value | <0.0001 | 0.055 | 0.467 | 0.000 |
| Df | 22 | 22 | 22 | 22 |

## 4. Discussion

In this study, the total abundance of syrphid flies was higher than the honey bees. *Apis dorsata* was the most abundant pollinator, followed by *E. aeneus*, while the lowest abundance was recorded for *I. scutellaris*. Previously, *A. dorsata* was reported as an abundant and efficient pollinator of different crops in Southern Punjab, i.e., onion [26], canola [18], bitter gourd [25], and pumpkin. Syrphid flies are characteristic of the spring season in the plains of semi-arid Punjab, Pakistan, as the spring season benefits them in terms of floral resources and weather conditions [26]. Contrarily, in subtropical Jammu of India, bees comprised about 90% of the total pollinator community visiting strawberry flowers, and the rest (10%) included syrphids and other insect visitor groups [15]. Across the world, honey bees (*A. mellifera*) and mason bees (*Osmia bicornis*) are used to provide stable pollination services in strawberry cultivation [2,6,13]. Bees are known to spread pollen homogeneously onto the receptacles of strawberry flowers, which results in an increased number of fertilized achenes [11,32]. Some syrphid fly species are also known to effectively pollinate strawberry flowers [6,10].

In our study, based on visitation rate and pollen harvest efficacy, *A. dorsata* and *E. aeneus* proved to be more efficient than *A. florea* and *E. corollae*. Visitation rate and pollen harvest are among the most important parameters for assessing the pollination efficiency of an insect pollinator [28,33,34]. Previously, from the study region, *A. dorsata* was reported as the best pollinator of canola, bitter gourd, and pumpkin in terms of visitation rate and pollen harvest [18,19,25]. Contrarily, stay time was the highest for *E. corollae*, while it was the lowest for *A. dorsata*. More time spent per flower is not beneficial to a plant as it leads to lower pollen dispersal, while greater pollinator movement (with lower stay time) promotes better flower outcrossing and plant reproductive success [35].

In the present study, open-pollination treatment resulted in heavier strawberry fruits, a greater fruit set ratio, and a reduced market maturity period as compared to the self-pollination treatment. Some previous studies have also reported 25–70% greater fruit set of marketable grade in honey-bee-pollinated strawberry fruits than caged fruits [2,6,10,15,16]. Some other studies suggest that syrphid flies not only improve the quality of strawberry fruits, i.e., fruit size, fruit weight percentage, fewer misshapen fruits [11,36], but also provide additional benefits in terms of biocontrol of aphids in the strawberry fields [10].

Since strawberry is a highly perishable fruit [37], its shelf life is among the most important quality parameter for consumers and the food industry. In the present study, open-pollinated fruits resulted in better shelf life (7 days) and marketable percentage (100% on 3rd day) than self-pollinated fruits (i.e., 4 days and 56% on 3rd day). Resemblant of many other fruits, i.e., oriental melons, cucumbers, and tomatoes [38,39], cross-pollination also improves the shelf life of strawberry fruits, as a previous study reported that open-pollinated strawberry fruits remained marketable up to the 5th day, while self-pollinated fruits remained marketable up to the 3rd day at room temperature [13]. The shelf life of strawberries depends upon their firmness, which is influenced by effective pollination. Auxin and gibberellic acid in healthy pollinated fruits delay fruit softening, leading to enhanced firmness and shelf life [40].

In the present study, open-pollinated fruits exhibited higher TSS (7%), TA (0.39), and vitamin C (220.6), and low pH (2.82) as compared to self-pollinated fruits. Furthermore, self-pollinated fruits had lower TSS (5%) and higher pH (3.8) than the recommended values. The effect of cross-pollination on the biochemical properties of strawberry fruits is poorly documented [13,15,16]. For marketable fruits, the value of TSS should be equal to 7%, while pH should be less than 3.7 [41].

Measuring pollinator effectiveness is a necessary first step towards understanding plant–pollinator interactions, especially in lesser-known biodiversity hotspots [42]. Determining single-visit effectiveness of pollinators in terms of seed or fruit set is more effective than pollen harvest or deposition on the stigma [22,24]. However, only a few studies have evaluated single-visit effectiveness of pollinators in terms of pollination rate (proportion of fertilized ovules) [6,43,44]. In our study, *A. dorsata* proved to be the most efficient pollinator

in terms of single-visit fruit set and fruit weight, followed by *E. aeneus*. The managed honey bees (*A. mellifera*) are regarded as better pollinators of strawberry in terms of pollination rate than the other native non-*Apis* bees [43,44]. However, another study did not find any significant difference in pollination rates in single visits of *A. mellifera*, Halictidae bees, and syrphid flies (*Eristalis* sp.), since the cultivar "Chandler" used in this study was less dependent on insect pollinators [6].

A recent study [8] reported heavier strawberry fruits from visits of wild bees than managed honey bees, although they did not find any significant differences in pollen deposition between both the groups. This difference in fruit weight was due to the delivery of more outcrossed pollen by wild bees than honey bees, as the pollen quality parameters (allogamous vs. geitonogamous or autogamous) are rarely quantified [45]. Wild bee *Apis dorsata* has has also been reported as the most efficient pollinator of other crops (in the study area) in terms of single-visit effectiveness, i.e., pumpkin, bitter gourd [25], canola [18], and onion [26].

The present study has three potential limitations. First, some other less abundant pollinator species were also observed in the focal plot, but were not included in our systematic counts. These less abundant species might outweigh the abundant species in their pollination effectiveness. Second, the study was performed on a limited scale, while pollinators vary spatially in their abundance and diversity. Third, the single-visit effectiveness of pollinators was not evaluated in terms of the biochemical characteristics of strawberry fruits, which otherwise can provide a better picture of their relative effectiveness than merely on the basis of physical characteristics.

In conclusion, *A. dorsata* and *E. aeneus* were the most efficient strawberry pollinators in terms of their single-visit efficacy. Moreover, cross-pollination in strawberry fruits always leads to better physical and biochemical characteristics. *Apis dorsata* is a well-known honey bee species—due to its honey—among the local farming communities [46], whereas *E. aeneus* is a poorly known species and is mostly ignored or even taken as an insect pest [47]. Conserving these pollinators can ensure sustainable strawberry production in the region. This can be achieved by increasing the awareness among farmers about identification and the role of pollinators through regular extension services; providing nesting sites for enhancing native insect pollinators [48]; using the least toxic insecticides at times when the activity of pollinators is low; preserving natural habitat surrounding the strawberry fields [49,50]; and ensuring the availability of diverse nectar and pollen resources across the year [51]. Keeping in view the highly urbanized landscape of the study site and the spatial variations in the abundance and diversity of pollinators, future studies should explore the single-visit effectiveness of a range of species on a landscape scale. Moreover, as well as physical characteristics, the biochemical characteristics of fruits should also be evaluated as a measure of single-visit effectiveness.

**Author Contributions:** All authors contributed to the study conception and design. Material preparation and data collection was conducted by M.A. (Muhammad Anees), while data analysis was performed by M.A. (Mudssar Ali). The first draft of the manuscript was written by M.A. (Muhammad Anees), and all the authors commented on subsequent versions of the manuscript. All authors have read and agreed to the published version of the manuscript.

**Funding:** The authors extend their appreciation to the Deanship of Scientific Research at King Khalid University Saudi Arabia for funding this work through Large Groups Project under grant number RGP.2/28/43.

**Acknowledgments:** We are grateful to Asghar Hassan (China Agricultural University, Beijing, China) for the identification of syrphid fly species.

**Conflicts of Interest:** The authors declare no conflict of interest.

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
