# Peer review of "Impact of Bee and Fly Pollination on Physical and Biochemical Properties of Strawberry Fruit"

_horticulturae, doi:10.3390/horticulturae8111072_

Round 1
Reviewer 1 Report
Manuscript is original, well defined, easy to understand. The language is appropriate and understandable. The topic is compatible with the journal’s scope. The results are very significant and relevant, presented in a well-structured manner. The manuscript’s results are reproducible based on the details given in the methods section. The figure (1) and tables (1 to 5) are appropriate, they are clearly presented. Conclusions justified and supported by the results, consistent with the evidence and arguments presented.
Accept manuscript with minor changes:
In the title of the manuscript for bee and fly add plural (bees and flies) due to the greater number of the species mentioned during the study impact on physical and biochemical properties of strawberry fruit.
Apstract:
Line 5: Apis florea abbreviate A. florea.
Line 10: Same as line 5.
Introduction.
Line 10: For the species Episyrphus balteatus the genus name and the species name (Episyphus baletatus) are misspelled.
Materials and Methods
Study area
Line 3: Write the Latin name of the strawberry species (var. Chandler).
Line 7 and line 8: wWhether a hyperlink is required?
Pollinator abundance
Line 5: Specify the reference according to the journal's propositions.
Results
Line 12: Apis dorsata abbreviate A. dorsata.
Discussion
Line 14: Without italics for word “and” (A. dorsata and E. aeneus).
Line 55: Apis dorsata abbreviate A. dorsata.
Line 66: Same as line 55.
Table 1: Add italics for species name (Apis dorsata).
Table 2: Write the name of the species correctly (E. corollae).
Author Response
Dear Anonymous Reviewer,
Thanks for improving the manuscript.
Best Regards

Reviewer 2 Report
Dear Authors,
Please find a manuscript - with my remarks - file attached.

Author Response

(The authors gave the same response as above.)

Reviewer 3 Report
This research is exciting to read, and the authors rightly identify the need to continue research in this field with this level of detail. Your efforts are to be commended.
There are a few comments that I would like to make to improve the overall quality and detail of the manuscript.
1) When discussing visual observations used to collect data, it would provide some authority to mention the possible differences between observers. This is commonly mentioned in research that relies on these methods. While this may not be necessary to include in your analyses, it would be good to mention in terms of caution in the interpretation of the results. This is likely a minor issue given the low species richness reported in the study.
2) There are many instances where data mentioned in tables do not match what is reported in the text. These clerical errors should be addressed.
3) For the single visit effectiveness and other instances where the flowers are caged. It would be helpful to understand what materials were used to create the cages. This would inform the readers how to replicate future studies and build on these data. It would also be more clear as to if the cages were open to wind pollination or much smaller insect pollinators.
4) You mention various biochemical parameters measured in the study. You should explain the abbreviations in the first instance used, not later in the methods section. This will help less familiar audiences. Furthermore, there is no mention of how many replicates are included in this portion of the data collection/analysis.
5) The results of the ANOVA analyses do not mention the denominator degrees of freedom. Furthermore, pole width is not significantly different p<0.190. This is either a clerical error, or some other misunderstanding (assuming alpha = 0.05).
6) P values should read p<0.000 not p=0.000. There is inconsistent formatting on this topic.
7) Results in a table format can be more difficult to comprehend the magnitude of the effect sizes you share. While there may be limitations to the number of figures, if you were able to add another figure regarding the pollen harvest or fruit set data, that would be helpful. I would suggest a box-and-whisker plot over a bar graph to show the distribution of the data rather than the mean and SE. I agree that the marketable fruit data is best reported as a figure given the added complexity to the data.
8) There are some inconsistent statements in the discussion section. You mention that TSS should be equal or less than 7% but mention the self-pollinated fruit had lower TSS (5%). This suggests that this is an undesirable value, which is contradictory to your previous claim. Furthermore, you include data comparing the Vitamin C values, which are not significantly different between the open and self-pollinating plants.
9) The sentence referencing Albano et al. 2009 in the discussion mentions the cultivar "Camerosa." It is unclear if you are referring to the cultivar used in Albano et al. 2009, or the "Chandler" cultivar used in this study. This sentence should be edited to resolve this confusion.
10) Finally, the paragraph beginning with, "A recent study,..." has many disconnected thoughts. This paragraph should be revised to reflect the relatedness of these ideas and its importance within the broader discussion.
Author Response

(The authors gave the same response as above.)
